# Comparison of the Kinematics Following Gait Perturbation in Individuals Who Did or Did Not Undergo Total Knee Replacement

**Vicktoria Elkarif [1], Leonid Kandel [2], Debbie Rand [1]** **, Isabella Schwartz [3,4], Alexander Greenberg [2], Rivkin Gurion [2] and Sigal Portnoy [1,***

[1] Department of Occupational Therapy, Sackler Faculty of Medicine, Tel Aviv University, Tel Aviv 6997801, Israel; Vicktor1@mail.tau.ac.il (V.E.); drand@tauex.tau.ac.il (D.R.)
[2] Department of Orthopaedics, Hadassah Medical Center, Mount Scopus, Jerusalem 9765418, Israel; kandel@hadassah.org.il (L.K.); alexander2@hadassah.org.il (A.G.); rivkingur@hadassah.org.il (R.G.)
[3] Faculty of Medicine, Hebrew University of Jerusalem, Jerusalem 9112001, Israel; isabellas@hadassah.org.il
[4] Hadassah Medical Center, Medicine & Rehabilitation, Orthopedic Department, Jerusalem 9765418, Israel
* Correspondence: portnoys@tauex.tau.ac.il

**Abstract:** We aimed to compare the spatiotemporal parameters and joint kinematics during unperturbed and perturbed gait between individuals with osteoarthritis (OA) who did or did not undergo total knee replacement (TKR) one year post a baseline evaluation. OA subjects scheduled for TKR (TKR group; $n = 14$) and not scheduled for TKR (NTKR group; $n = 17$) were age-matched. Outcome measures included: joint range of motion, timed up and go, joint pain levels, Oxford score, and the Activities-specific Balance Confidence Scale. In addition, spatiotemporal gait parameters and joint kinematics were recorded during perturbed and unperturbed gait. After one year, most of the TKR group (71%), but only 41% of the NTKR group, increased their gait velocity by more than 0.1m/sec, which is the meaningful clinical important difference for gait velocity. After perturbation of the contralateral limb, the TKR group showed a greater decrease in the maximal extension of the OA hip compared to the NTKR group ($p = 0.031$). After perturbation of the OA limb, more subjects decreased their OA knee flexion–extension range in the NTKR group compared to the TKR group ($p = 0.011$) and more subjects decreased their maximal ankle plantar flexion in the TKR group ($p = 0.049$). Although the surgery was successful in terms of pain reduction and increased functionality, individuals following TKR exhibited unique compensatory strategies in response to the perturbation of both limbs. These findings might suggest that balance deficits remain in individuals following TKR and therefore are associated with a risk of falls.

**Keywords:** gait analysis; arthroplasty; fall; knee surgery; joint pain; gait perturbation





## 1. Introduction

Osteoarthritis (OA) is a degenerative disease that is comprised of biological, structural, and biomechanical components [1–4]. Individuals with OA may suffer joint deformities, pain, and muscle weakness [3]. As the severity of OA increases, proprioceptive accuracy, i.e., the accuracy of the sense of position and movement of joints or extremity in space, decreases. Consequently the joint stabilization is compromised [5]. Poor proprioception is often associated with poor muscle strength and functional limitation [6,7]. Consequently, poor proprioception, joint deformity, and muscle weakness are among the contributing factors of falling in individuals with OA [7–11]. Thus, eventually, OA may lead to physical impairment and functional limitations, e.g., gait abnormality [2–4]. Several gait parameters have been found to be predictive of falls [12]. For example, asymmetry of the double support duration was a strong predictor of repeated falls in elderly women [13]. Abnormality of the spatiotemporal gait parameters such as gait velocity [14,15], swing [15], and stance

durations, in addition to step length [14] and toe clearance [15,16] were correlated with the incidence of falls. Another factor that may affect the outcome of tripping is the strategy of recovery following perturbation [17]. However, this possible factor was not thoroughly explored in individuals with OA, specifically those who underwent surgical treatment.

Total knee replacement (TKR) is often required in advanced OA when there is a significant decrease in functionality and after conservative interventions have failed. The prevalence of TKR surgery increases every year [18]. TKR reduces pain, improves functionality and quality of life [19], increases gait velocity [20], and range of motion (ROM) [21]. While these are encouraging outcomes of the surgery, individuals that undergo TKR remain at a high risk of falls [10]. Previous literature showed differences in the gait parameters between individuals following TKR and non-OA groups [22]. In addition, gait biomechanics were compared between individuals with OA before TKR and OA groups that were not scheduled for TKR [23,24]. Unfortunately, there are no studies that objectively quantified the post-surgery biomechanical effects over a long duration after TKR compared to individuals with OA who were not scheduled for TKR. In addition, most studies concerning fall strategies of individuals with knee OA were limited to perturbations applied during standing [25] or while stepping onto a translating platform [26]. To the best of our knowledge, no studies explored the recovery from a perturbation in individuals with knee OA (following TKR or not following TKR) while walking on a paved path. The gait stability, characterized by the ability to keep functional locomotion following a perturbation or contact with an obstacle [27], depends on the reactive recovery responses [28], e.g., compensatory stepping or counter rotation, in order to keep the projection of the center of gravity inside the base of support [29,30]. Therefore, the response of the first step following gait perturbation is meaningful and is utilized for the destabilizing effects after loss of balance [17,23,28]. In individuals with OA, the response to perturbation might be abnormal due to muscle weakness [31,32], pain [31], joint rigidity, decreased ROM [33], or decreased proprioception [34].

As insight into the changes of biomechanics following gait perturbation of individuals following TKR over a long duration may shed light on the existing factors that contribute to the risk of falls in those populations, we aimed to compare the gait characteristics during unperturbed and perturbed gait in individuals with OA who did or did not undergo TKR one-year post a baseline evaluation.

## 2. Materials and Methods

### 2.1. Population

We recruited 37 individuals from the Orthopedics clinic at the Hadassah Medical Center. The inclusion criteria were: 60–80-year-olds with knee pain, diagnosed with OA by orthopedic surgeons, and able to walk 5 m without assistive devices. The exclusion criteria were: hip, ankle, or contralateral (CL) knee disorders (OA diagnosis for the CL limb or above 5 in report of pain in the Visual Analogue Scale; VAS); previous lower limb surgery; systemic joint disease; and neurological or vestibular impairment. Individuals who underwent surgery of the lower limbs after the recruitment, aside from the planned TKR, were removed from the study. The criteria used by the orthopedic surgeons to warrant surgery relied on the severity of the radiographic findings and the physical examination, as well as the subjective pain severity levels, functional disability, and no or limited progress assessment following rehabilitation. Individuals that did not undergo TKR were recruited for this study as a control group, as there are no data regarding the recovery strategy following gait perturbation in this population. Therefore, the control group is important to understand the effect of the TKR on the primary outcome measures of this study. Candidates were approached after their consultation with the physician, who did or did not refer them to surgery to ensure that the selection of the group of subjects who were referred or not referred to TKR was not biased on their participation in this study. The study recruitment process and design are depicted in Figure 1. The study was approved

by the hospital's Helsinki committee (approval #0045-15-HMO). All the study procedures were performed according to the Declaration of Helsinki.

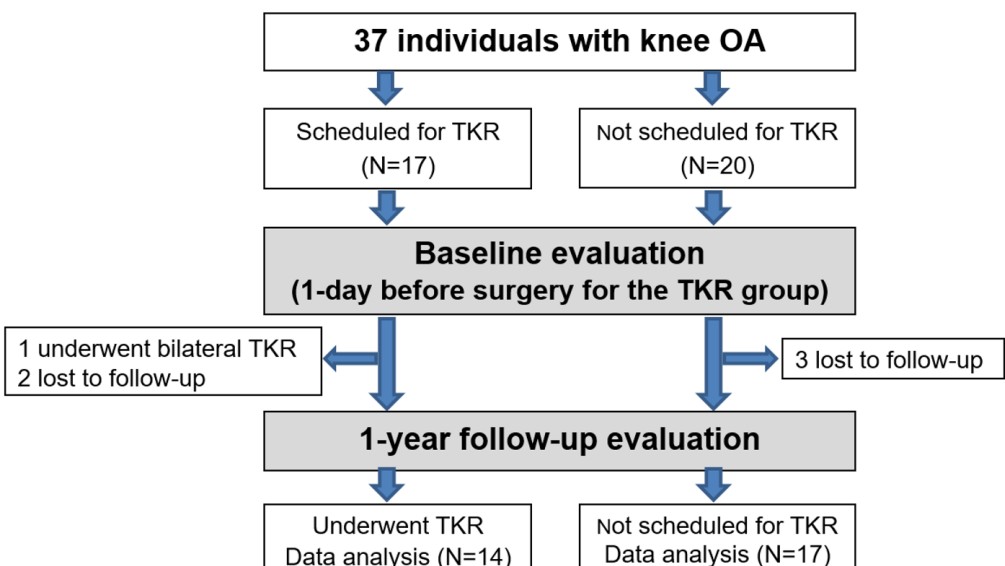

**Figure 1.** The study recruitment process and design.

### 2.2. Tools

The subjects filled out a demographic questionnaire. Passive ROM of the hip and knee was measured bilaterally. The timed up and go (TUG) was used as a predictor of the risk of fall [35]. The minimal detectable change (MDC) of TUG is 1.1 s [36].

Kinematics were collected using a 10-camera motion capture system (Qualisys, Sweden). A ceiling-attached safety harness was used to prevent falls. A manually-controlled electronic locking mechanism was attached to the ankle by a velcro strap attached to an unstretchable cable (Figure 2). The cable went through a wall-mounted pulley, set at one end of the path at ankle height. Locking of the pulley was controlled by a mechanism based on a relay shield [23]. When pressing a button, the cable is locked and then released after a duration of 200 ms, causing a short gait perturbation. The locking mechanism therefore produced an immediate and complete stop for a duration of 200 ms, as if the limb contacted a rigid obstacle in its path. This effect was similar between subjects walking at different velocities.

We used three questionnaires: First, the subjective pain levels were assessed using the VAS, rated from '0' (no pain) to '10' (worst imaginable pain) in both knees in the last week. A change of ≥20 mm is the established meaningful clinical important difference (MCID) for pain in individuals with knee OA [37]. The MCID represents the smallest improvement considered worthwhile by a patient. Second, the Oxford score for functional disability, which consists of 12 questions (scored between '0' meaning low disability to '4' meaning high disability) concerning the pain and disability experienced over the past four weeks. A change of ≥7 points is the established meaningful important difference (MID) [38]. Third, we used the Activities-specific Balance Confidence (ABC) Scale to assess the fear of falling. A score below 67% in older adults predicted the risk for falling [39].

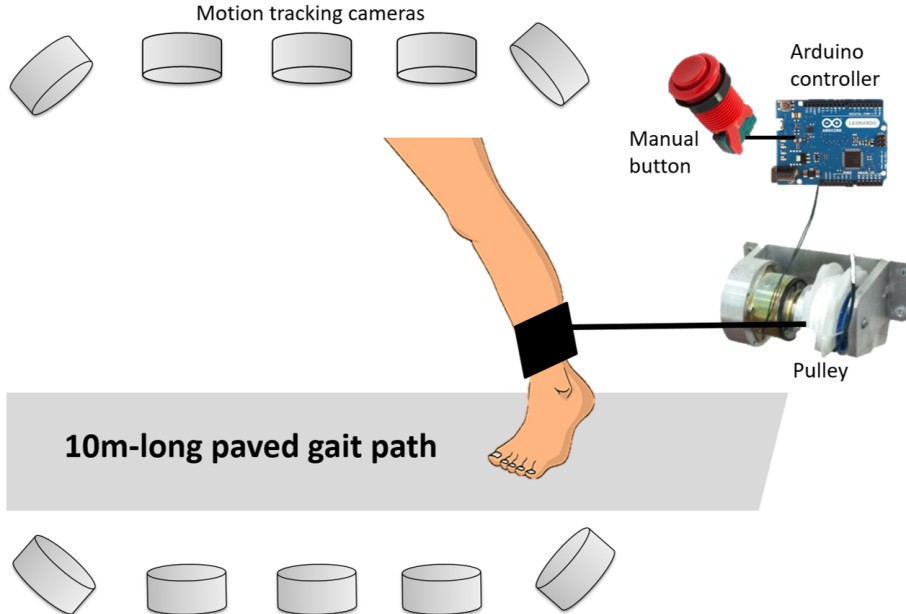

**Figure 2.** The subjects walked on a 10 m-long paved path surrounded by 10 motion-tracking cameras. A manually-controlled electronic locking mechanism was attached to the ankle by a strap. A cable from the strap is released from a wall-mounted pulley, set at one end of a 10 m path at ankle height. Locking of the pulley was controlled by a mechanism based on a relay shield, an Arduino controller, and a manual button. When pressing a button, the cable is locked and then released after a duration of 200 ms, causing a short gait perturbation.

### 2.3. Protocol

Each subject signed an informed consent form and was tested twice: at baseline (for the TKR, one day pre-surgery) and at follow-up one year after the baseline session. In each session, physical and functional measures were collected and the subject filled out the three questionnaires. Twenty markers were placed on the subject at anatomical locations: on the posterior and anterior superior iliac spines of the pelvis; the medial and lateral femoral condyles of the distal femur; the medial and lateral malleoli; the posterior tip of the calcaneus; the base of the second metatarsal; and the head of the first and fifth metatarsi. In addition, four clusters, which included four markers each, were attached to the lateral aspects of the hip and calf. The subject walked three times on a 10 m paved path at a comfortable speed with a safety harness. Then, the instrumented cable was attached above the ankle of either limb. The subject walked again at least three times on the same path during which two random perturbations were induced in mid-swing for each leg. The subject was asked to keep walking following the perturbation.

### 2.4. Post-Analysis

We used the Qualisys Track Manager software to extract the markers' 3D coordinates. For the unperturbed gait, the initial contact and toe-off of several strides were manually-marked. For the perturbed gait, we analyzed the data of the first perturbation of each limb.

The markers' coordinates and timings were exported to a custom code created in LabView (v2017, National Instruments, Austin, TX, USA). We calculated the gait velocity, cadence, stance and swing durations, and the Symmetry Index (SI) for the step length and double support durations of the unperturbed gait according to:

$$SI = \frac{|X_R - X_L|}{(X_R + X_L) \cdot \frac{1}{2}} \cdot 100 \tag{1}$$

where $X_R$ and $X_L$ are the values of a spatial and temporal parameter of the right/left leg, respectively. For the joint kinematics calculations, we used a commercial software (Visual

3D, C-Motion, Germantown, MD, USA). For the pain, ROM, and questionnaire scores, we calculated the difference between the follow-up test and the baseline test. For the gait and perturbation data, we calculated the percentage of change between the two trials as follows:

$$X\,[\%] = \frac{X_{Follow-up} - X_{Baseline}}{X_{Baseline}} \cdot 100 \tag{2}$$

The statistical analysis was performed in SPSS v23. The normal distribution was tested using the Shapiro–Wilk test. Normal data are presented as average and standard deviation (SD), and data that were not normally distributed are presented as the median and interquartile range (IQR). The unpaired *t*-test or Mann–Whitney test were used to compare between the two groups. Statistical significance level was $p < 0.05$.

## 3. Results

The demographic characteristics, physical and functional measurements, and pain reports are summarized in Table 1. There were no between-group differences in all of the aforementioned parameters.

**Table 1.** Demographic characteristics, difference in physical and functional measurements, and pain reports of the two groups calculated by: $X_{follow\ up} - X_{Baseline}$. Numeric values are presented as the median and interquartile percentage.

| | Variable | NTKR (*n* = 17) | TKR (*n* = 14) | *p* | t |
|---|---|---|---|---|---|
| | Sex | 7 male; 10 female | 4 male; 10 female | 0.473 | −0.718 |
| | Age (years) | 68.3 (7.3) | 70.8 (6.4) | 0.586 | −0.997 |
| | BMI (kg/m²) | 29.1 (4.7) | 30.8 (4.4) | 0.687 | −0.994 |
| | Injured knee | 6 left; 11 right | 8 left; 6 right | 0.231 | −1.197 |
| Functional measurements | OXFORD (0–48) | −6.9 (29.1) | −31.2 (46.5) | 0.164 | 1.774 |
| | ABC (0–100%) | −4.9 (38.4) | 1.7 (38.3) | 0.845 | −0.471 |
| | TUG (s) | −11.6 (17.6) | −12.6 (27.8) | 0.078 | 0.116 |
| VAS of pain | Knee OA limb | 0.4 (2.6) | −3.8 (3.3) | 0.281 | 3.984 |
| | Knee CL limb | 1.4 (3.7)) | 1.0 (4.1) | 0.986 | 0.276 |
| Passive Range of Motion (°) | Hip flexion OA limb * | 0.0 (0.0–0.0) | 0.0 (0.0–2.5) | 0.404 | −0.843 |
| | Hip flexion CL limb * | 0.0 (0.0–0.0) | 0.0 (0.0–0.0) | 0.598 | −0.527 |
| | Hip extension injured limb * | 0.0 (0.0–0.0) | 0.0 (0.0–0.0) | 0.420 | −0.807 |
| | Hip extension CL limb * | 0.0 (0.0–0.0) | 0.0 (0.0–0.0) | 0.485 | −0.699 |
| | Knee flexion OA limb | 0.0 (13.0) | 4.6 (16.3) | 0.296 | −0.882 |
| | Knee flexion CL limb * | 0.0 (−10.0–0.0) | 0.0 (−5.0–5.0) | 0.290 | −1.057 |
| | Knee extension OA limb * | −2.5 (0.0–2.5) | 0.0 (0.0–1.3) | 0.414 | −0.817 |
| | Knee extension CL limb * | 0.0 (0.0–0.0) | 0.0 (0.0–0.0) | 0.163 | −1.395 |

* As the data were not normally distributed, values are shown as the median and interquartile percentage, and the Mann–Whitney test was used (Z value is presented instead of t). Abbreviations: BMI = Body Mass Index: $25 \le$ overweight $< 30$, $30 \le$ obese $< 35$, $35 \le$ obesity; VAS = Visual Analogue Scale; TUG = timed up and go; ABC = Activities-specific Balance Confidence Scale; CL = contralateral; OA = osteoarthritis; CL = contralateral; TKR = total knee replacement; and NTKR = no TKR.

In the TKR group, 11 (78%) subjects reported a decrease of more than 20 mm in the VAS score of the OA limb one year following the TKR, while in the NTKR group, only three (11%) reported a decrease of more than 20 mm in the VAS score of the OA limb. In the OXFORD score, 10 (71%) TKR subjects from the group reported a decrease of more than seven points one year following the TKR, while in the NTKR group, only two (10%) reported a decrease of more than seven points. Eight (57%) TKR subjects decreased their TUG score by more than 1.1 s one year following the TKR, while in the NTKR group, eight (47%) decreased the TUG score by more than 1.1 s. Finally, eight (47%) NTKR subjects and six (42%) TKR subjects had ABC scores lower than 67%.

The percentage of change of the spatiotemporal and joint kinematics data of the unperturbed gait are summarized in Table 2. The percentage of change of the swing duration of the CL limb was lower in the NTKR group compared to the TKR group ($p = 0.038$). There were no other between-group differences in the spatial–temporal gait

parameters. However, after one year, ten (71%) TKR subjects and seven (41%) NTKR subjects increased their gait velocity by more than 0.1 m/sec, which is the MCID [40].

**Table 2.** Spatiotemporal and kinematics data of the lower limbs of the unperturbed gait calculated by $X[\%] = \frac{X_{Follow-up} - X_{Baseline}}{X_{Baseline}} \cdot 100$. Numeric values are presented as the average and standard deviation.

| | Parameter | NTKR (*n* = 17) | TKR (*n* = 14) | *p* | t |
|---|---|---|---|---|---|
| Spatial–temporal parameters | Velocity (m/s) * | 5.8 (−5.4–20.2) | 17.8 (9.9–44.8) | 0.104 | −1.625 |
| | Cadence (steps/min) | 1.4 (6.9) | 2.8 (10.3) | 0.152 | −0.409 |
| | Swing duration (s) of CL limb | −2.4 (7.2) | −1.0 (10.6) | 0.038 | −0.373 |
| | Swing duration (s) of OA limb * | −5.3 (−10.0–3.0) | −4.6 (−8.0–8.5) | 0.368 | −0.900 |
| | Symmetry index step length | −33.8 (41.9) | −19.2 (64.3) | 0.092 | −0.660 |
| | Symmetry index of double support * | 21.2 (−35.6–244.6) | −28.9 (−43.2–120.5) | 0.161 | −1.401 |
| Kinematics | Pelvic maximal posterior tilt | −35.7 (97.7) | −148.9 (196.9) | 0.032 | 1.733 |
| | OA hip range of flexion–extension * | 62.1 (9.0–234.8) | 48.1 (4.4–120.5) | 0.651 | −0.453 |
| | CL hip range of flexion–extension | 3.3 (27.3) | −23.0 (50.3) | 0.041 | 1.760 |
| | OA knee range of flexion–extension | 8.0 (22.5) | 18.0 (46.2) | 0.006 | −0.721 |
| | CL knee range of flexion–extension * | 36.6 (−3.5–370.8) | 39.7 (−8.4–322.0) | 0.806 | −0.245 |

* As the data were not normally distributed, values are shown as the median and interquartile percentage, and the Mann–Whitney test was used (Z value is presented instead of t). Abbreviations: OA = osteoarthritis; CL = contralateral; TKR = total knee replacement; and NTKR = no TKR. The Symmetry Index was calculated according to: *Symmetry index* $= \frac{|X_R - X_L|}{(X_R + X_L) \cdot \frac{1}{2}} \cdot 100$, where $X_R$ and $X_L$ are the values of a spatial or temporal parameter of the right/left leg, respectively.

Three between-group differences were found for joint kinematics. The percentage of reduction in the maximal posterior pelvic tilt during the OA limb step was significantly higher in the TKR group compared to the NTKR group (*p* = 0.032). In addition, the percentage of change of the flexion–extension range of the hip in the CL limb was significantly higher in the TKR group compared to the NTKR group (*p* = 0.041). Lastly, the percentage of change of the knee flexion–extension range for the OA limb was higher in the TKR group compared to the NTKR group (*p* = 0.006).

The percentage of change of the perturbed gait characteristics are summarized in Table 3. There were no significant differences between the two groups in the percentage of change between the baseline and the follow-up tests for all spatial–temporal parameters. As for the kinematics, when the CL limb was perturbed, the percentage of reduction in the maximal OA hip extension was higher in the TKR group compared to the NTKR group (*p* = 0.031). When the OA limb was perturbed, the percentage of change in the maximal flexion–extension range of the OA knee was lower in the TKR group compared to the NTKR group (*p* = 0.011). In addition, there was a marginal difference in the percentage of change of the maximal ankle plantar flexion (*p* = 0.049).

**Table 3.** Spatiotemporal and kinematics data of the lower limb of the perturbed gait immediately after the perturbation and the first step following the perturbation calculated by $X[\%] = \frac{X_{Follow-up} - X_{Baseline}}{X_{Baseline}} \cdot 100$. Numeric values are presented as the average and standard deviation.

| Parameter | NTKR (*n* = 17) | TKR (*n* = 14) | *p* | t |
|---|---|---|---|---|
| *Contralateral limb tripped* | | | | |
| Stance duration | 91.4 (76.1–102.3) | 91.1 (74.6–126.0) | 0.312 | 1.030 |
| Step length | 19.4 (4.8–38.6) | 22.2 (9.8–38.2) | 0.188 | −1.356 |
| Base width | 89.1 (75.6–146.2) | 130.6 (98.5–160.6) | 0.989 | 0.014 |
| CL hip maximal extension * | −89.9 (−110.1–(−37.4)) | −106.5 (−157.8–(−88.4)) | 0.323 | −0.989 |
| OA hip maximal extension | 19.4 (161.9) | −234.8 (252.7) | 0.031 | −2.937 |
| CL knee maximal flexion * | −9.5 (−49.4–53.8) | −18.7 (−843.7–29.6) | 0.569 | −0.570 |
| OA knee maximal flexion | −3.0 (34.2) | −11.9 (43.7) | 0.244 | 0.563 |
| *Injured limb tripped* | | | | |
| Stance duration | 77.9 (73.6–116.9) | 85.3 (75.8–139.9) | 0.219 | −1.260 |
| Step length | 15.7 (6.5–77.1) | 41.3 (16.5–169.6) | 0.069 | 1.996 |
| Base width | 110.8 (66.7–142.2) | 97.2 (76.3–142.7) | 0.167 | −1.430 |

**Table 3.** *Cont.*

| Parameter | NTKR (*n*= 17) | TKR (*n* = 14) | *p* | t |
|---|---|---|---|---|
| | Injured limb tripped | | | |
| CL knee range of flexion–extension * | 101.6 (11.7–408.2) | 80.2 (5.0–443.2) | 0.958 | −0.053 |
| OA knee range of flexion–extension | −7.8 (41.3) | −5.9 (39.9) | 0.011 | −0.988 |
| Maximal OA ankle plantar flexion | −34.0 (36.5) | −35.9 (17.2) | 0.049 | −0.157 |
| Maximal CL ankle plantar flexion * | 4.1 (−19.7–61.3) | 6.7 (−9.0–89.9) | 0.770 | −0.293 |

Abbreviations: OA = osteoarthritis; CL = contralateral; TKR = total knee replacement; and NTKR = no TKR * As the data were not normally distributed, values are shown as the median and interquartile percentage, and the Mann–Whitney test was used (Z value is presented instead of t).

## 4. Discussion

We compared the gait patterns during a unperturbed and perturbed gait in individuals with OA who did or did not undergo TKR one year post a baseline evaluation. This is the first report of gait characteristics and perturbation change in the recovery strategy of subjects with OA (who did or did not undergo TKR) over one year. Our main findings show a decrease in pain levels and Oxford scores in most subjects following TKR, coupled with increased gait velocity and reduced pelvic tilt, as well as a different perturbation recovery strategy compared to the NTKR group.

Although there were no between-group differences in the change in VAS and Oxford scores, when considering the MCID of the Oxford score and the VAS pain score of the OA limb, it is evident that the majority of TKR subjects showed improvement in these scores, while the majority of NTKR subjects did not show improvement. These findings imply that the surgery was successful in reducing the pain and disability levels, and thereby may improve the functionality of the subjects. As for the ABC and TUG, approximately half of the subjects in both groups had a reduced TUG score by more than the MDC and had an ABC score lower than the cut-off for risk of falling. These results may indicate that following one year, the fear and risk of falling were similar between individuals with knee OA who did or did not undergo surgery. Thus, although the gait parameters following TKR are improved, they might not predict the risk of falls, which still occur following TKR [10]. This might be attributed to worsening in knee proprioception [41], balance deficiency [31], or residual characteristics of pre-surgery gait [42].

A systemic review argued that gait velocity correlates with functional abilities in adults with mobility deficiencies [43]. In our study, most of the TKR subjects improved their gait velocity by at least 0.1 m/sec compared to less than a half in the NTKR group. It is therefore not surprising that most TKR subjects improved their Oxford scores, while only 10% of the NTKR subjects showed improvement. Another association of the gait velocity concerns joint kinematics. Several studies, e.g., [44], reported correlation between decreased velocity and a decrease in knee/hip flexion angles. Most of the TKR subjects increased the ROM of the OA knee during unperturbed gait. A pathology of one knee may affect the entire dynamics of all joints in both lower limbs [45]. For example, walking with a stiff knee results in decreased ROM in the CL limb [46]. In our study, most of the TKR subjects increased the ROM of their OA knee following the surgery, and in most subjects in this group, the ROM of the CL hip and the posterior pelvic tilt were decreased. The aforementioned effect found in the TKR group is not apparent in the NTKR group. The greater increase in ROM of the OA knee in the TKR group might be attributed to the reduction in pain following the surgery [47]. The lack of pain reduction in most NTKR subjects might also explain the greater reduction in swing duration in the CL limb one year post baseline, which might imply that individuals in the NTKR group increased their avoidance of weight-bearing on their OA limb. We previously reported [23] that the gait ROM of the OA limb was significantly lower in individuals scheduled for TKR compared to individuals not-scheduled for TKR. In addition, the ROM of the OA limb of both OA groups was lower than the ROM of the knees of healthy controls [23]. The findings of the current study therefore imply that the ROM of the OA knee of the TKR subjects was more similar to healthy controls [23] one year post-surgery. These changes seen in the TKR

group, but not in the NTKR group, may be the key to clinical improvement of gait velocity and functionality in the TKR group.

Two significant differences were found following the perturbation of the OA limb: more NTKR subjects decreased their OA knee flexion–extension range but more TKR subjects decreased their maximal ankle plantar flexion. These findings suggest that the kinematics of the NTKR group move away from the perturbation recovery strategy of healthy adults [23]. Conversely, the recovery strategy of the TKR group trends towards the recovery strategy of healthy adults [23]. The differences in change in recovery strategies of both groups in regards to that of healthy adults [23] might be explained by muscle weakness or joint stiffness around the knee due to the OA or secondary to surgery. Furthermore, the decrease in maximal ankle plantar flexion might result from co-contraction of the Gastrocnemius and Tibialis anterior before the initial contact of the perturbed limb which limits balance recovery, thereby increasing the risk of falling. Individuals after TKA rely on the non-operated limb, thereby exhibiting asymmetrical gait [46]. This was evident as TKR subjects decreased their OA hip extension when the CL limb was perturbed. This might result from latent activation of the muscles surrounding the knee or adjacent joints. Another explanation for the non-normal recovery strategy is the loss of proprioception due to OA [34] or the artificial prosthesis. Although the surgery aims to correct joint alignment [48], it is unable to compensate for proprioception insufficiencies and might lead to difficulties in responding to perturbation. These explanations were previously mentioned in literature concerning the dynamic balance during gait in individuals post-TKR [29,49].

The study limitations include, firstly, the small single-center sample size that might not generalize for the entire OA population. Secondly, a learning effect might have been introduced following the first perturbation. Finally, we did not monitor ground reaction forces that might have added to our understanding of the response to perturbation.

To conclude, although the surgery was successful in reducing pain and increasing functionality, individuals following TKR exhibited asymmetrical gait and unique compensatory strategies in response to perturbation. These findings might suggest that balance deficits remain in individuals following TKR and therefore are associated with the risk of falls. It might therefore be hypothesized that an optimal rehabilitation program following TKR that includes gait balance exercises will improve gait symmetry, balance recovery after perturbation, and reduce falls.

**Author Contributions:** Conceptualization, V.E., L.K. and S.P.; methodology, V.E., L.K., D.R., I.S. and S.P.; software, V.E. and S.P.; validation, V.E. and S.P.; formal analysis, V.E. and S.P.; investigation, V.E. and S.P.; resources, I.S., L.K., A.G. and R.G.; data curation, V.E.; writing—original draft preparation, V.E. and S.P.; writing—review and editing, V.E., L.K., D.R., I.S. and S.P.; visualization, V.E. and S.P.; supervision, S.P. and D.R.; project administration, I.S., L.K. and A.G. All authors have read and agreed to the published version of the manuscript.

**Funding:** This study received no external funding.

**Institutional Review Board Statement:** The study was approved by the Hadassah hospital's Helsinki committee (approval #0045-15-HMO).

**Informed Consent Statement:** Informed consent was obtained from all subjects involved in the study.

**Conflicts of Interest:** The authors declare no conflict of interest.

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
