# Peer review of "Comparison of the Kinematics Following Gait Perturbation in Individuals Who Did or Did Not Undergo Total Knee Replacement"

_applsci, doi:10.3390/app11167453_

Round 1

Reviewer 1 Report

The paper shows interesting and valuable results of kinematic comparison of gait under perturbation. these results are useful for gait assessment of individuals, particularly for patients with OA and TKR and also the elderly. 

the paper is well written and organized. however, some details are missing, as listed below.

  1. introduction. the review of relevant work in both assessment methods and results is too short and insufficient.
  2. method. whether subjects walked on the floor, or treadmill? Fig. 2 shows only gait perturbation. A whole view of the setup is desirable.
  3. data are missing in the paper. plots of spatiotemporal parameters are recommended.

in their setting, a strap connected by a cable is used for perturbation. the cable used is elastic or unstretchable? what is the influence of the cable type on the gait? 

in the work, it is only motion capture system is used. why forceplates are not used. please discuss.

please use fonts consistently throughout the paper.

Reviewer 2 Report

General Comments

The authors investigate the response of OA patients treated with TKA, and not treated with TKA to an induced gait perturbation one-year post-operative and one year after an initial examination respectively. The authors do not describe how the recruited patients were selected or randomized into the respective (TKA, and non TKA) groups. Furthermore, the authors seemed to have forgotten to mention at which center the patients were recruited, and have instead just inserting a place holder „xxxxxx Medical Center“ in the manuscript. Everyone can make a mistake, but this does not speak much for the diligence of the authors in the preparation of the manuscript. Furthermore, without the reader knowing how the patients were selected for each of the two study groups, it is impossible for the reviewer to judge the scientific merit of this study. There are several potential sources of bias between the patients deselected for TKA that could make the comparison meaningless. It should also be considered, that the pathology of non TKA patients will progress in the year up to the follow-up exam, and there may be no sense in comparing them to the post TKA cohort. The benefits of TKA are proven (see the gait velocity results of the authors), and if the non TKA patients have significant pain it might not be ethically justifiable to request them to hold out and endure the pain for another year; this could for example result in an increased risk of opioid addiction for those patients. Another issue is the conclusion in the Abstract; the authors where there is no mention the non TKA group, and only the TKA group is referred to. This raises further issues about necessity of the non TKA group. The authors should consider addressing these issues and having an editor or colleague read the paper to scan for simple mistakes before resubmitting the work to a scientific journal.

Round 2

Reviewer 1 Report

all comments by this reviewer were addressed in the revsion. the reviewer does not have further comments.

Author Response

We thank the reviewer for taking the time to review and approve our revision.